# Simulations of the Characteristics of the Entropy Mode in Dipole-Magnetic-Confined Plasmas

**DOI:** 10.3390/e25111481

**Published:** 2023-10-26

**Authors:** Liang Qian, Zhibin Wang, Jian Chen, Aohua Mao, Yi Yv, Qiuyue Nie, Xiaogang Wang

**Affiliations:** 1Department of Physics, Harbin Institute of Technology, Harbin 150001, China; lqian@hit.edu.cn (L.Q.); aohuamao@hit.edu.cn (A.M.); xgwang@hit.edu.cn (X.W.); 2Sino-French Institute of Nuclear Engineering and Technology, Sun Yat-sen University, Zhuhai 519082, China; 3Laboratory for Space Environment and Physical Sciences, Harbin Institute of Technology, Harbin 150001, China; 4School of Electrical Engineering and Automation, Harbin Institute of Technology, Harbin 150001, China

**Keywords:** entropy mode, dipole-magnetic-confined plasmas, simulation, space plasma

## Abstract

Plasmas confined in a dipole magnetic field widely exist in both space and laboratories, and this kind of plasma draws much attention from researchers both in plasma physics and in space science. In this paper, the characteristics of the collisionless electrostatic instability of the entropy mode in a dipole-magnetic-confined plasma are simulated with the linear gyrokinetic model. It is found that the entropy mode can be generated in dipole-magnetic-confined plasmas, and there are two typical stages of the entropy mode, with another transitional stage at different values of *η*. The main instability changes from the ion diamagnetic drift to the electronic diamagnetic drift as *η* becomes larger. In addition, the MHD mode predicts that the most stable point is at *η*~2/3 when *k*_⊥_*ρ*_i_ << 1. However, we find that *η* and *k*_⊥_*ρ*_i_ are coupled with each other, and the most stable point of the mode moves gradually to *η*~1 as *k*_⊥_*ρ*_i_ increases. There is a peak value for the entropy mode growth rate around *k*_⊥_*ρ*_i_~1.0, and more complicated modes are induced so that the dispersion relation has been changed when the driving force of the plasma pressure gradient effect is obvious. For example, the characteristics of the interchange-like modes gradually emerge when the driving effect of the plasma pressure becomes stronger. Further investigations should be taken to reveal the characteristics of the entropy mode in magnetospheric plasmas.

## 1. Introduction

The earth’s magnetic field (or the simulated earth’s magnetic field configuration in the laboratory) is a typical dipole magnetic field, which can be regarded as a “magnetic mirror” with a strong magnetic field at the two poles and a weak magnetic field at the middle equatorial plane. These kinds of configurations have a good constraint effect on the plasmas confined in the magnetic field [1,2,3]. Since the dipole magnetic field is a kind of magnetic field configuration that is relatively easy to design and implement in the laboratory, plasma confined by a dipole magnetic field is often used in laboratories to carry out scientific research on plasma characteristics under magnetic field confinement conditions [4,5,6,7]. At present, the dipole-magnetic-field-confined plasma research devices under construction (or that have been built) include the SPERF (Space Plasma Environment Research Facility) at Habin Institute of Technology in China, the CTX (Collisionless Terrella eXperiment) at Columbia University in USA, the LDX (Levitated Dipole eXperiment) at MIT in the USA, and the RT-1 (Ring Trap-1) at Tokyo University in Japan, etc. [8,9,10,11].

In the dipole magnetic field, the charged particles bounce back at the magnetic mirror point and perform a reciprocating bouncing motion when they move toward the two poles. The curvature drift, gradient drift, and bouncing motion of charged particles, together with the spiral motion of electrons, constitute the basic motion form of electron capture in a dipolar field; Figure 1 shows the motion of a charged particle in the radiation belt. Theoretically, the ideal dipole magnetic field has a good confinement property for the charged particles, but collisions and turbulence disturbances will enhance particle transport and destroy the confinement. Different from tokamaks, which have both good curvature regions and bad curvature regions, there is no shear field in the dipole magnetic confined plasmas, and the whole region holds the feather of bad curvature. Therefore, the plasma compression plays a key role in the MHD interchange instability in the dipole-magnetic-confined plasmas. Recent studies show that the plasma equilibrium in the dipole magnetic field has good MHD stability properties [12,13,14,15].

In fact, not all the modes can be described by the MHD equations, especially for the unstable modes which do not satisfy the MHD ordering assumption. In the case of a weak gradient, the MHD modes can be stabilized; however, a large number of non-MHD modes have been developed, and the entropy mode is a typical one. The entropy mode was first studied by Kadomtsev in 1960 [16]. When the interchange mode in the plasma is considered, the plasma and the flux tube move together, and the inside flux tube and outside flux tube exchange their positions as well as the plasma contained in the flux tube. The interchange modes are usually treated as ideal MHD, and an adiabatic equation of state *pv*^γ^ = *c* is generally adopted. In this case, the entropy of the plasmas does not change while exchanging their positions. The free energy is derived from the total pressure gradient of the plasma. For the entropy mode studied in this paper, the electrons and the ions propagate in opposite directions, and each has its own density disturbance, temperature disturbance, and pressure disturbance. However, there is no total pressure disturbance in the leading order, which leads to the changes in the specific entropy of the plasmas. The driven source of this mode can be the density and/or temperature gradients, rather than the total pressure gradient.

Recently, the entropy mode in the dipole magnetic field was investigated by many researchers using fluid models and kinetic models [17,18,19,20,21]. Furthermore, the entropy mode can also be observed in laboratories [17,22,23]. The influences of the temperature anisotropy and the multicomponents on the entropy mode are also included [24,25]. However, the basic assumption of these studies is that the entropy mode has flutelike instability, that is, *k*_‖_ ≈ 0, where *k*_‖_~*m*/*R*_c_, and the bounce average is adopted to simplify the model. It is known that the Landau damping has a strong stabilizing effect unless *k*_‖_*v*_ti_ << *ω* << *k*_‖_*v*_te_ for the entropy mode with the frequency *ω*~ < *ω_D_* > *_θ_*~*k*_⊥_*ρ*_i_*v*_t_/*R*_c_, where *R*_C_ is the radius of curvature. It can be seen that the finite *k*_‖_ has a strong stabilizing effect when *k*_⊥_*ρ*_i_ << 1, that is, the instability in this range is mainly a flutelike instability. However, the parallel dynamic plays an important role for the plasma instability when *k*_⊥_*ρ*_i_ > 1. This destroys the conservation of the second adiabatic invariant J of the particle, which makes the bounce average ineffective.

In this paper, linear gyrokinetic simulations with parallel dynamics are taken to analyze the behaviour of the entropy mode in a dipole magnetic field configuration. The layout of this paper is as follows: Section 1 is the introduction, and the model used in this paper is described in Section 2. Simulation results of the characteristics of the entropy mode in the typical dipole-magnetic-field-confined plasma are presented in Section 3. Finally, this paper is summarized in Section 4 with concluding remarks.

## 2. Model Description

The Maxwell equation should be satisfied for any magnetic field configuration:(1)∇⋅B=0
(2)$$\nabla \times B=\frac{1}{c^{2}}\frac{\partial E}{\partial t}+\mu_{0}J$$
where ***B*** is the magnetic field intensity, ***E*** is the electric field intensity, *c* is the light velocity, *t* is the time, ***J*** is the current density, and *μ*_0_ is the vacuum magnetic permeability. The equilibrium background field is considered in Equation (2), and the timescale of the evolution is much larger than the disturbance timescale for the equilibrium field considered in this paper. Therefore, the background magnetic field can approximately be regarded as not evolving with time within the disturbance timescale. It is often assumed that the system has reached a steady state and there is no current in that region when considering the distribution of the background equilibrium fields. Therefore, Equation (2) can be rewritten as
(3)∇×B=0

When the curl of a physical quantity is zero, it can be written as the gradient of a physical quantity. Assuming that the physical quantity is *Ψ*(*r*), we can obtain
(4)$$\nabla \cdot B=-\nabla^{2}\Psi \left(r\right)=0$$

Equation (4) can be solved by separating variables in a spherical coordinate system,
(5)Ψ(r)=R(r)Θ(θ)Φ(φ)
where *R*, *θ*, and *Φ* can be expressed as follows:(6)$$R\left(r\right)=A_{n}^{}r^{n}+\frac{B_{n}^{}}{r^{n+1}}$$
(7)$$\Phi \left(\varphi \right)=C_{m}^{}\cos\left(m\varphi \right)+D_{m}^{}\sin\left(m\varphi \right)$$
(8)$$\Theta \left(\theta \right)=E_{n}^{m}P_{n}^{m}\left(\cos\theta \right)$$
where $P_{n}^{m}$ is the associated Legendre polynomial, and *A_n_*, *B_n_*, *C_m_*, *D_m_*, and $E_{n}^{m}$ are the undetermined coefficients. Considering that the magnetic field is zero at infinity, therefore *A* is 0, and the solution to the equation can be written as
(9)$$\Psi =\sum_{n=1}^{\infty \;}\sum_{m=0}^{n}\frac{1}{r^{n+1}}P_{n}^{m}\left(\cos\theta \right)\left(C_{n}^{m}\sin\left(m\varphi \right)+D_{n}^{m}\cos\left(m\varphi \right)\right)$$

When the lowest order approximation is *m* = 0 and *n* = 1, the potential generated by Equation (9) is the ideal dipole field potential. In such a dipole field, the components of the magnetic field can be expressed as
(10)$$B_{r}=-2B_{0}\frac{R_{E}^{3}}{r^{3}}\cos\theta$$
(11)$$B_{\theta }=-B_{0}\frac{R_{E}^{3}}{r^{3}}\sin\theta$$
where *B*_0_ represents the magnetic field intensity at the equatorial surface. It can be seen that due to the cyclic symmetry of the dipole field, there is no φ component. The above equations are usually represented by the magnetic moment *M* = $B_{0}R_{E}^{3}$ (the unit is Am^2^). However, the above representation method is very complex in the calculation because the calculation demand will be greatly increased in the analysis, and the simulation process as the background equilibrium cannot be well expressed by the magnetic field. The plasma equilibrium is required, which means that the macroscopic thermodynamic parameters (such as the pressure, the temperature, the number density, etc.) are almost the same as those on the magnetic surface, and the magnetic surface coordinate system is often more advantageous for calculation.

For the dipole field potential generated above, the magnetic plane coordinate system (*ψ*, *χ*, *ζ*) can be used; thus, it is obtained that
(12)$$\left\{\begin{array}{c} \psi =\frac{Msin^{2}\theta }{r}\\ \chi =\frac{Mcos\theta }{r^{2}}\;\\ \zeta =\varphi \end{array}\right.$$
where *M* is the magnetic moment at the equator of the dipole field. The magnetic plane coordinate system selected in this way is orthogonal and satisfies B=∇φ×∇ψ=∇χ, and *B*=$\sqrt{B\cdot B}$ = (*M*/*r*^3^)$\sqrt{1+3\cos^{2}\theta }$, which is very convenient for calculation.

The Maxwell distribution function *F*_0s_ = *n*_0s_*F*_M_ is used, where *n* is the number density. $F_{M}={\left(\frac{m}{2\pi T}\right)}^{3/2}\exp\left(-\frac{m\varepsilon }{T}\right)$ is the Maxwellian distribution, where 0 represents the equilibrium quantity, and *s* represents the species of the particles. For equilibrium plasma, *F*_0_ = *F*_0_(*ψ*), namely, the balance quantities are only the function of the magnetic surface *ψ*.

Based on the gyrokinetic theory, the particle perturbation distribution function with the gyrophase average can be expressed as follows:(13)$$f_{s}=\frac{q_{s}}{m_{s}}\frac{\partial F_{0s}}{\partial \epsilon }\phi +J_{0}\left(k_{⊥\rho_{s}}\right)h_{s}$$
where *q* stands for the electric charge of the species, *ϕ* is the electrical potential, *J*_0_ is the zero-order Bessel function, and *h*_s_ is the nonadiabatic gyrokinetic response. *v* is the velocity and *ε* = *v*^2^/2 is the particle energy. Compared to the flat plate model, the dipole field configuration has the magnetic drift effect of the charged particles. In this configuration, the time evolution of the nonadiabatic disturbance distribution function of charged particles is listed as follows:(14)$$\left(\omega -\omega_{Ds}+iv_{\|}b\cdot \nabla \right)h_{s}=-\left(\omega -\omega_{*s}^{T}\right)\frac{\partial F_{0s}}{\partial \epsilon }\frac{q_{s}}{m_{s}}J_{0}\phi$$

In this model, the plasma is assumed to be collisionless, which is the same as those in the geospace. Certain parameters mentioned above are listed as follows:(15)$$\rho_{s}=\frac{v_{⊥}}{\Omega_{s}},\;\Omega_{s}=\frac{q_{s}B}{m_{s}c},\;\mu =\frac{v_{⊥}^{2}}{2B},\;\frac{\partial F_{s}}{\partial \epsilon }=-\frac{m_{s}}{T_{s}}F_{s},\;b=B/B.$$
(16)$$\omega_{*s}^{T}=\frac{k_{⊥}\times b\cdot \nabla F_{0}^{s}}{-\Omega_{s}F_{0\varepsilon }^{s}},\;\omega_{Ds}=k_{⊥}\cdot v_{d}=k_{⊥}\cdot b\times \frac{\mu \nabla B+v_{\|}^{2}b\cdot \nabla b}{\Omega_{s}}$$
where $\omega_{*s}^{T}$ and *ω*_Ds_ are the antimagnetic drift frequency and magnetic drift frequency and *b* is the unit vector along the magnetic field, respectively.

The long drift wave assumption is adopted, namely, *kλ*_D_ << 1, where *λ*_D_ is the Debye length. It is generally believed that under such assumptions, the collective mode effect of the plasma is more dominant rather than the two-body collisions. Therefore, the quasi-neutral condition is used to replace the Poisson equation to obtain the closure equation of the model,
(17)$$\sum_{s}q_{s}\int f_{s}d^{3}v=0$$
where *f*_s_ stands for the gyrokinetic distribution function of the species.

It is defined that
(18)$$g_{s}\equiv h_{s}-\frac{q_{s}}{T_{s}}F_{0}^{s}J_{0}\phi$$

In the one-dimensional simulation along the magnetic field line, $b\cdot \nabla =\partial_{l}$. Therefore, Equation (14) can be rewritten as [26]:(19)$$\left(\partial_{t}+v_{\|}\partial_{l}\right)g_{s}=-i\omega_{Ds}g_{s}-i\left(\omega_{Ds}-\omega_{*s}^{T}\right)\;\frac{q_{s}}{T_{s\;}}F_{0}^{s}J_{0}\phi -v_{\|}\frac{q_{s}}{T_{s\;}}F_{0}^{s}\left[J_{0}\partial_{l}\phi -J_{1}\partial_{l}\left(k_{⊥}\rho_{s}\right)\phi \right]$$

The Bessel function *J*ʹ_0_ = −*J*_1_ is used, and for the electrons and ions with only one component, the quasi-neutral condition can be rewritten as
(20)$$-\frac{q_{i}n_{i}}{T_{i}}\left(1-\Gamma_{0i}\right)\phi +\int J_{0i}g_{i}d^{3}v=-\frac{q_{e}n_{e}}{T_{e}}\left(1-\Gamma_{0e}\right)\phi +\int J_{0e}g_{e}d^{3}v$$

There is $dl=r_{0}\kappa (\xi )d\xi$ along the magnetic field line; thus, it is obtained that
(21)$$\frac{\partial }{\partial l}=\frac{\partial }{\partial \xi }\frac{\partial \xi }{\partial l}=\frac{1}{r_{0}\kappa \left(\xi \right)}\frac{\partial }{\partial \xi }$$
where *ξ* = π/2 − *θ*. The motion of the charged particles along the magnetic field line is $dl=v_{\|}dt$, and the evolution equation of the particle position is obtained as follows:(22)$$\frac{d\xi }{dt}=\frac{v_{\|}}{r_{0}\kappa }$$

During the particle motion, $v_{\|}$ and $v_{⊥}$ change with its position along the magnetic field line, but *ϵ* and *μ* are conserved along the magnetic field line. Defining the pitch angle of the equatorial plane λ ≡ *μB*_0_/*ϵ*, it is obtained that $\left|v_{\|}\right|=\sqrt{2(\epsilon -\mu B)}=v\sqrt{1-\lambda B/B_{0}}$, where $v=\sqrt{2\epsilon }$, $v^{2}=v_{\|}^{2}+v_{⊥}^{2}$ and the magnetic field $B=B_{0}f(\xi )$. Therefore, Equation (30) can be rewritten as
(23)$$\frac{d\xi }{dt}=\frac{\pm v\sqrt{y}}{r_{0}\kappa }$$
where y=1−f(ξ)λ. In addition, under the action of magnetic mirror force, the velocity of the charged particles changes, and the velocity equation is
(24)$$\frac{dv_{\|}}{dt}=-\frac{\mu \nabla_{\|}B}{m}=-\frac{\lambda v^{2}}{2}\frac{df}{r_{0}\kappa d\xi }$$

The magnetic drift velocity ***v***_g_, ***v***_c_, caused by the magnetic field gradient and the curvature, and the total drift velocity ***v***_d_ can be expressed as
(25)$$v_{g}=\frac{1}{m\Omega_{s}}\mu b\times \nabla B=-\frac{v_{⊥}^{2}}{2\Omega_{s}B}\left(b\times \nabla B\right){\hat{e}}_{\phi }$$
(26)$$v_{c}=\frac{1}{\Omega_{s}}v_{\|}^{2}\nabla \times b=-\frac{v_{\|}^{2}}{\Omega_{s}B}\left(b\times \nabla B\right){\hat{e}}_{\phi }$$
(27)$$v_{d}=-\frac{b\times \nabla B}{B\Omega_{s}}\left(v_{\|}^{2}+\frac{1}{2}v_{⊥}^{2}\right){\hat{e}}_{\phi }=-\frac{g}{r_{0}\Omega_{s0}f}\left(v_{\|}^{2}+\frac{1}{2}v_{⊥}^{2}\right){\hat{e}}_{\phi }$$
where $g(\xi )=r_{0}(b\times \nabla B)/B$, $\nabla \times b=\nabla \times \frac{B}{B}=\frac{1}{B}\nabla \times B+\left(\nabla \frac{1}{B}\right)\times B=\frac{b}{B}\times \nabla B$, and ∇×B=0.

In addition, the perpendicular wave vector is
(28)$$k_{⊥}^{2}=k_{\psi }^{2}\nabla \psi \cdot \nabla \psi +k_{\zeta }^{2}\nabla \zeta \cdot \nabla \zeta =\frac{k_{p}^{2}}{r_{0}^{2}p{\left(\psi \right)}^{2}}+\frac{k_{t}^{2}}{r_{0}^{2}p{\left(\zeta \right)}^{2}}$$
and the gyro-radius is
(29)$$\rho_{s}=\frac{v_{⊥}}{\Omega_{s}}=\frac{v\sqrt{\lambda }}{\Omega_{s0\;}}\frac{1}{\sqrt{f\left(\xi \right)}}$$

The final magnetic drift frequency obtained is
(30)$$\omega_{Ds}=k_{⊥}\cdot v_{d}=-\frac{k_{t}g}{r_{0}^{2}\Omega_{s0}f_{p}}\left(v_{\|}^{2}+\frac{1}{2}v_{⊥}^{2}\right)=-\frac{\omega_{d0}g}{fp_{\zeta }}\frac{\left(1+y\right)}{2}v^{2}$$
where $\omega_{d0}=k_{t}/(r_{0}\Omega_{s0})$. In a shear-free dipole magnetic field configuration, there are
(31)$$\begin{array}{cc} \left(k_{⊥}\times b\right)\cdot \nabla n_{0} & =\frac{1}{B}\left[\left(k_{\psi }\nabla \psi +k_{\zeta }\nabla \zeta \right)\times \nabla \chi \right]\cdot \nabla \psi \frac{\partial n_{0}}{\partial \psi }\\ & =\frac{1}{B}\left[k_{\zeta }\nabla \zeta \times \nabla \chi \right]\cdot \nabla \psi \frac{\partial n_{0}}{\partial \psi }=k_{\zeta }B\frac{\partial n_{0}}{\partial \psi } \end{array}$$
and the gradient diamagnetic drift frequency is
(32)$$\begin{array}{cc} \omega_{*s}^{T} & =\frac{k_{⊥}\times b\cdot \nabla F_{0}^{s}}{-\Omega_{s}F_{0\epsilon }^{S}}=\frac{k_{⊥}\times b\cdot \nabla F_{0}^{s}}{m_{s}\Omega_{s}F_{0}^{s}/T_{s}}\\ & =\frac{k_{\zeta }B\frac{\partial n_{0}}{\partial \psi }\frac{1}{n_{0}}\left\{1+\eta_{s}\left[\frac{m_{\epsilon }}{T_{0}}-\frac{3}{2}\right]\right\}}{m_{s}\Omega_{s}/T_{s}}\\ & =-\omega_{*s}\left\{1+\eta_{s}\left[\frac{m_{\epsilon }}{T_{0}}-\frac{3}{2}\right]\right\} \end{array}$$
where $\omega_{*s}=k_{t}cT_{s}/(q_{s}L_{ns})$ and $L_{n}\equiv -{\left(\frac{\partial \ln n_{0}}{\partial \psi }\right)}^{-1}$.

The normalization scale is chosen for the ion thermal velocity *v*_ti_, and the radius *r*_0_ at *ξ* = 0. The standard *δf* method is used, the particle weight is defined to be *w_s_* = *g_s_*/*F*_0s_, and the normalized evolution equation can be expressed as follows [26]:(33)$$\frac{d\xi_{s}}{dt}=\frac{v_{\|}}{\kappa }$$
(34)$$\frac{d\omega_{s}}{dt}=-i\omega_{Ds}\omega_{s}-i\left(\omega_{Ds}-\omega_{*s}^{T}\right)\frac{q_{s}}{T_{s}}J_{0}\phi -v_{\|}\frac{1}{\kappa }\frac{q_{s}}{T_{s}}\left[J_{0}\partial_{\xi }\phi -J_{1}\partial_{\xi }\left(k_{⊥}\rho_{s}\right)\phi \right]$$
(35)$$\left(1+\frac{1}{\tau_{e}}-\Gamma_{0i}-\frac{1}{\tau_{e}}\Gamma_{0e}\right)\phi =\int J_{0i}g_{i}d^{3}v-\int J_{0e}g_{e}d^{3}v$$
(36)$$\frac{dv_{\|}}{dt}=-\frac{v^{2}\lambda }{2}\frac{1}{\kappa }\frac{df}{d\xi }$$

The governing Equations (34)–(36) are the equations simulated by the PIC method. For any magnetic field, the corresponding drift instability can be obtained with *f*(*ξ*), *κ*(*ξ*), *g*(*ξ*), *p*(*ψ*), and *p*(*ζ*). In an ideal dipole field, $\kappa =\cos\xi {\left(1+3\sin^{2}\xi \right)}^{1/2}$, $f(\xi )\frac{\sqrt{1+3\sin^{2}\xi }}{\cos^{6}\xi }$, $p(\psi )=B_{0}r_{0}^{2}\sqrt{1+3\sin^{2}\xi }/\cos^{3}\xi$, and $p(\zeta )=1/\cos^{3}\xi$.

The normalized parameters are
(37)$$\frac{dv_{\|}}{dt}=-\frac{v^{2}\lambda }{2}\frac{3\sin\xi \left(3+5\sin^{2}\xi \right)}{\cos^{8}\xi (1+3\sin^{2}\xi )}$$
(38)$$k_{⊥}^{2}=\left(k_{\zeta }^{2}+k_{\psi }^{2}z\right)/\cos^{6}\xi$$
(39)$$k_{⊥}\rho_{s}=\frac{v}{v_{ts}}\sqrt{\lambda }\sqrt{\left(k_{\zeta }^{2}+k_{\psi }^{2}z\right)z^{-1/4}\rho_{ts}}$$
(40)$$\partial_{\xi }\left(k_{⊥}\rho_{s}\right)=\frac{v}{v_{ts}}\sqrt{\lambda }\frac{\left(-k_{\zeta }^{2}+k_{\psi }^{2}z\right)}{2z^{5/4}\sqrt{k_{\zeta }^{2}+k_{\psi }^{2}z}\;}3sin\xi \cos\xi \rho_{ts}$$
(41)$$b_{s}=\rho_{ts}^{2}\left[k_{\zeta }^{2}+k_{\psi }^{2}z\right]\frac{\cos^{6}\xi }{z}$$
(42)$$\omega_{Ds}=-\omega_{ds0}\frac{3\left(1-\sin^{4}\xi \right)}{{\left(1+3\sin^{2}\xi \right)}^{2}}\frac{\left(1+y\right)}{2}\frac{v^{2}}{v_{ts}^{2}}$$
where $z=1+3\sin^{2}\xi$, and $k_{\psi }\to k\psi B_{0}r_{0}^{2}$.

## 3. Simulation Results and Discussion

Based on the linear gyrokinetic model and the PIC method mentioned above, numerical simulations are taken to reveal the basic characteristics of the entropy mode in the dipole-magnetic-field-confined plasmas. The influences of three important parameters on the entropy mode are investigated in this paper. Firstly, we focus on the ratio of the temperature gradient to the density gradient at typical *k*_⊥_*ρ*_i_, which represents the relative intensity between the temperature gradient and the density gradient. The curves of the mode frequency *ω*_r_ and the growth rate of the mode *ω*_i_ with different *η* = *L*_n_/*L*_T_, where $L_{T}\equiv -{\left(\frac{\partial \ln T}{\partial \psi }\right)}^{-1}$, are shown in Figure 2, where the values of *k*_⊥_*ρ*_i_ are set to be 1.0 and 1.5, respectively. It can be seen from Figure 2a that there is a rapid ascent stage at *η* = 0.7~0.9 for *k*_⊥_*ρ*_i_ = 1.0, and the rapid ascent stage is at *η* = 0.7~1.2 for *k*_⊥_*ρ*_i_ = 1.5, which is a broader range for the value of *η*. In Figure 2b, there are also two extreme points for the growth rate of the modes at adjacent points, for example, *η* = 0.8 for *k*_⊥_*ρ*_i_ = 1.0 and *η* = 0.9 for *k*_⊥_*ρ*_i_ = 1.5, instead of *η*~0.67 predicted by Ref. [27]. It is indicated in Figure 2 that there are two typical stages of the entropy modes, with a turning point at the value of *η* around 0.7. The stage with smaller *η* is dominated by the ion diamagnetic drift, while the stage with larger *η* is dominated by the electronic diamagnetic drift. A transitional stage connects the two typical stages, which exist at the median values of *η.* This is also consistent with the current theory [27]. However, it is found that *η* and *k*_⊥_*ρ*_i_ are coupled with each other, and the most stable point of the mode moves gradually to *η*~1 as *k*_⊥_*ρ*_i_ increases, which has not been noticed before.

The curves of the mode frequency *ω*_r_ and the growth rate of the mode *ω*_i_, with different *η* at *k*_⊥_*ρ*_i_ = 0.1 under typical 1/*L*_P_ are shown in Figure 3, where 1/*L*_P_ = −1/*P*(∂ln*P*_0_/∂*Ψ*). It can be seen that the curves exhibit different curve morphologies for the growth rate subfigure (Figure 3b), especially at smaller *η* where the number density gradient dominates. At a small value of *η*, the characteristics of the entropy mode are shown when the value of 1/*L*_P_ is not large enough, which is driven both by the number density gradient and the temperature gradient. However, when the value of 1/*L*_P_ becomes larger, the larger values of the mode growth rate are obtained, and the characteristics of the interchange-like modes gradually emerge. For example, the mode growth rates at 1/*L*_P_ = 6 and 8 in Figure 3b are large enough to enter the interchange-like mode.

In this paper, we focus on the stage with smaller *η* that is dominated by the ion diamagnetic drift, and it is shown that the entropy mode gradually transitions to the interchange-like mode when the driving effect of the plasma pressure becomes stronger (as 1/*L*_P_ increases), while for the mode frequency *ω*_r_, there are small differences in the curve morphology at different values of 1/*L*_P_, as shown in Figure 3a. At a larger value of *η*, the driving force of the temperature becomes stronger, and there may be other kinds of modes (such as drift wave modes), which are much more complicated to analyze.

The curves of the mode frequency *ω*_r_ and the growth rate of the mode *ω*_i_, with different 1/*L*_n_ at *k*_⊥_*ρ*_i_ = 0.5 under typical 1/*L*_T_ are shown in Figure 4a,b, respectively, where 1/*L*_T_ = −1/*n*(d*n*/d*Ψ*) and 1/*L*_T_ = −1/*T*(d*T*/d*Ψ*). It can be seen that at small 1/*L*_T_ (such as 1/*L*_T_ = 0 and 1/*L*_T_ = 1 in Figure 4) with a smaller temperature gradient, the curves of the mode frequency and the growth rate are quite similar, which is at the typical entropy mode stage at a small density gradient. However, the driving force of the plasma temperature is added with the increase in the value of 1/*L*_T_, and complicated modes are introduced in this system so that the entropy mode is therefore not the dominant mode. For example, there are different kinds of modes for the case of 1/*L*_T_ = 5 in Figure 4, which is consistent with the results shown in Figure 3.

The dispersion relations between *k*_⊥_*ρ*_i_ and the mode frequency *ω*_r_ and the dispersion relations between *k*_⊥_*ρ*_i_ and the growth rate of the mode *ω*_i_ are shown in Figure 5a,b, respectively, where *k*_⊥_*ρ*_i_ = 0.5 and *η* = 0.2. it is indicated that for the smaller value of *L*_p_^−1^ (for example, *L*_p_^−1^ = 5), the driving force of the plasma pressure gradient effect is not obvious, and the characteristics of the entropy mode are obtained, where there is a peak value of the mode growth rate around *k*_⊥_*ρ*_i_~1.0 that is consistent with the theory. While the driving force of the plasma pressure gradient effect is obvious for the larger value of *L*_p_^−1^ (for example, *L*_p_^−1^ = 10), more complicated modes are induced so that the dispersion relation has been changed totally.

## 4. Conclusions

In this paper, simulations on the characteristics of the entropy mode in a dipole-magnetic-field-confined plasma are taken based on a gyrokinetic model. The major influences of the typical parameters are revealed, and the main conclusions of this paper are summarized as follows:(1)It is found that the entropy mode can be generated in dipole-magnetic-confined plasmas, and there are two typical stages of the entropy mode, with another transitional stage at different values of *η*. The main instability changes from the ion diamagnetic drift to the electronic diamagnetic drift as *η* becomes larger and *η*~1 as *k*_⊥_*ρ*_i_ increases.(2)For the case with small values of *k*_⊥_*ρ*_i_ and *η*, the characteristics of the entropy mode are shown when the value of 1/*L*_P_ is small. However, the characteristics of the interchange-like modes gradually emerge when the driving effect of the plasma pressure becomes stronger (the value of 1/*L*_P_ becomes larger).(3)There is a peak value for the entropy mode growth rate around *k*_⊥_*ρ*_i_~1.0, and more complicated modes are induced so that the dispersion relation has been changed when the driving force of the plasma pressure gradient effect is obvious.

Further investigations should be taken on the entropy mode with dipole-magnetic-confined plasma to reveal the characteristics of the entropy mode in magnetospheric plasmas.

## Figures and Tables

**Figure 1 entropy-25-01481-f001:**
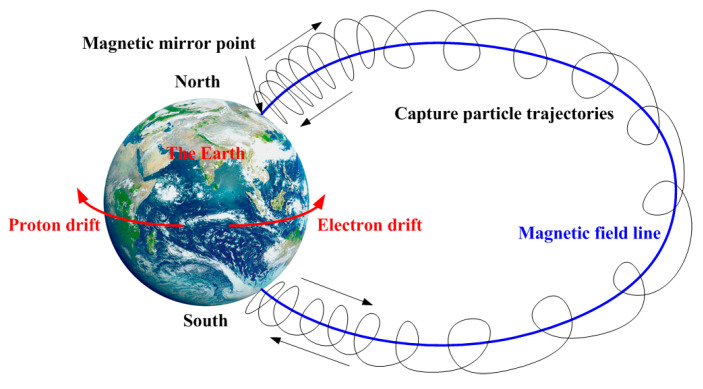
Schematic diagram of the basic motion of capture particles in the radiation belt.

**Figure 2 entropy-25-01481-f002:**
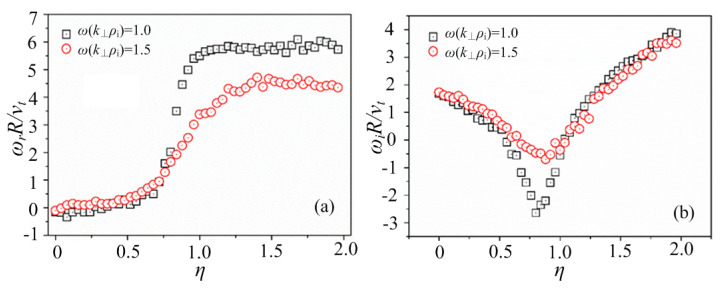
The curves of (**a**) the mode frequency *ω*_r_ and (**b**) the growth rate of the mode *ω*_i_, with different *η* = *L*_n_/*L*_T_ at typical *k*_⊥_*ρ*_i_.

**Figure 3 entropy-25-01481-f003:**
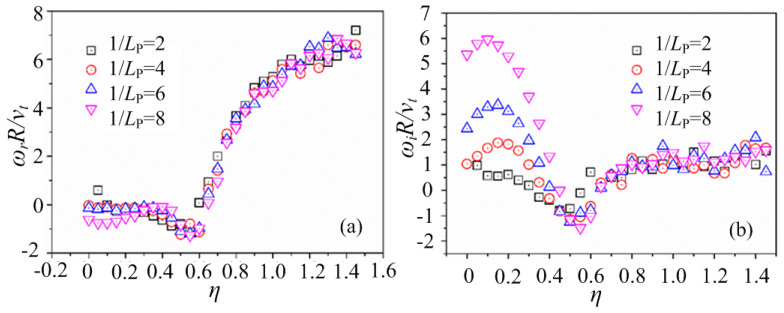
The curves of (**a**) the mode frequency *ω*_r_ and (**b**) the growth rate of the mode *ω*_i_, with different *η* = *L*_n_/*L*_T_ at the typical value of *L*_p_^−1^, where *k*_⊥_*ρ*_i_ = 0.1.

**Figure 4 entropy-25-01481-f004:**
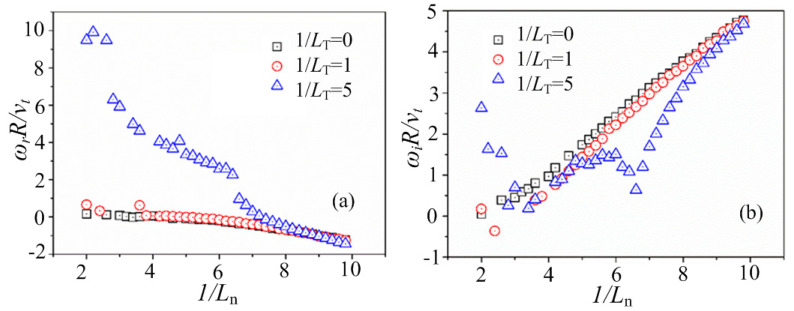
The curves of (**a**) the mode frequency *ω*_r_ and (**b**) the growth rate of the mode *ω*_i_, with different 1/*L*_n_ = −1/*n*(d*n*/d*Ψ*) at the typical value of *L*_T_^−1^, where *k*_⊥_*ρ*_i_ = 0.5, 1/*L*_T_ = −1/*T*(d*T*/d*Ψ*), and *η* = 0.2.

**Figure 5 entropy-25-01481-f005:**
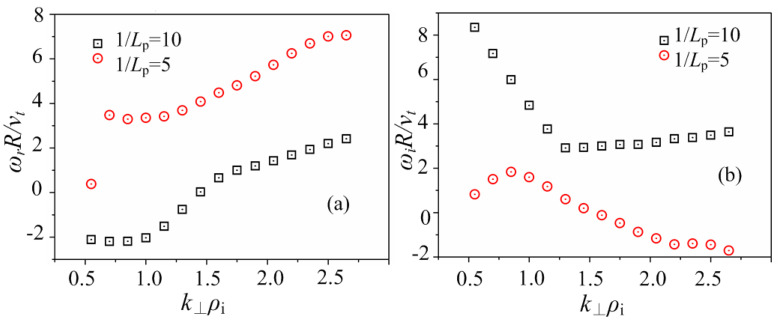
The dispersion relations between *k*_⊥_*ρ*_i_ and (**a**) the mode frequency *ω*_r_, and (**b**) the growth rate of the mode *ω*_i_, for different values of *L*_p_^−1^, where *k*_⊥_*ρ*_i_ = 0.5 and *η* = 0.2.

## Data Availability

The data that support the findings of this study are available from the corresponding author upon reasonable request.

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
