# Peer review of "Simulations of the Characteristics of the Entropy Mode in Dipole-Magnetic-Confined Plasmas"

_entropy, 2023, doi:10.3390/e25111481_

Round 1
Reviewer 1 Report
Please see attached file.

The grammar should be improved, as there are some badly constructed sentences, and some typos.
Reviewer 2 Report
The manuscript provides a very detailed description of the theory and method of examining the entropy mode in a dipole magnetic field. PIC simulation is used to obtain precise values of wave frequency and growth rate in various parameter regimes. This is a very exhaustive effort and the wave characteristics at different values of parameters are clearly determined. The results are important achievements of this area of research. I highly recommend its publication without need for changes.
Author Response
Thank the Referee for agreeing with the content of this paper and the recommendation for publication!
Reviewer 3 Report
Dear Authors, although I acknowledge the non negligible amount of work you have done, at the moment I find some difficulty to assess the suitability of your manuscript for publication. Beside of some technical remarks I have, my main element of concern is about the element of novelty of your work, which so far I have not been able to fully appreciate.
I therefore invite you to take account of my remarks for possible amendments, and I especially invite you to revise the introduction and structure of the manuscript, so to convey more clearly the novelty and importance of your results, with respect to previous literature.
Here my remarks.
- Abstract, Introduction and Conclusions: Please express more clearly what advancement on the subject your work aims to bring, by also expanding the referenced bibliography. By reading the text —and also from the title— it seems to me that the main element of novelty on which your work focusses, is the study of entropy modes in a dipole configuration. In this regard I note however there is a number of related works, which seem to me relevant, but which have not been cited. They should be added to the references and your results put in context with respect to them (also note that Kobayashi et al (2010) seems to me to have already pointed out the dominance of entropy modes in a dipole geometry at k? ∼ 1).
- Simakov, A. N., Catto, P. J., & Hastie, R. J. (2001). Kinetic stability of electrostatic plasma modes in a dipolar magnetic field. Physics of Plasmas, 8(10), 4414-4426.
- Levitt, Ben, Dimitri Maslovsky, and Michael E. Mauel. "Measurement of the global structure of interchange modes driven by energetic electrons trapped in a magnetic dipole." Physics of Plasmas 9.6 (2002): 2507-2517.
- Garnier, D. T., Boxer, A. C., Ellsworth, J. L., Hansen, A. K., Karim, I., Kesner, J., ... & Roach, A. (2008). Stabilization of a low-frequency instability in a dipole plasma. Journal of plasma physics, 74(6), 733-740.
- Kobayashi, S., Rogers, B. N., & Dorland, W. (2009). Gyrokinetic simulations of turbulent transport in a ring dipole plasma. Physical review letters, 103(5), 055003.
- Kobayashi, S., Rogers, B. N., & Dorland, W. (2010). Particle pinch in gyrokinetic simulations of closed field-line systems. Physical review letters, 105(23), 235004.
- Ou, W., Wang, L., Li, B., & Rogers, B. N. (2020). Turbulent pinch in whole-plasma simulations of a dipole-confined plasma. Physical Review E, 101(2), 021201.
- Qian, T. M., & Mauel, M. E. (2020). Observation of weakly damped modes using high resolution measurement of turbulence in a dipole confined plasma. Physics of Plasmas, 27(1).
- Abstract and Introduction: Also, I invite you to better highlight in the text what elements of novelty your present results are bringing with respect to your previously published work
- Qian, L., Wang, Z., & Wang, X. (2019). Gyrokinetic investigations on entropy modes in dipole magnetic field confined plasmas with an anisotropic temperature. Physics of Plasmas, 26(3).
- Introduction, lines 43-77: You devote a large part of the Introduction (almost the whole page 2 of the manuscript) to discuss the importance of pressure anisotropy, both in the geometry configurations you are going to study and for entropy modes. Later on, however, it seems to me that you completely neglect it, without saying a word about the reason for which you make such a choice. Why? In my opinion you could considerably shorten this section, by just mentioning that entropy can be important but you neglect it for some reason (for the sake of simplicity of the model?) in the present work.
- Introduction, lines 44-49: If you want to keep this part about the important of temperature anisotropy in plasma physics, I notice that your discussion is quite limited to some specific applications : there are lot of topics in which this subject may enter, with a huge related bibliography that goes well beyond the couple of references [10,11] which you have mentioned (for example: the generation of anisotropic distributions due to cyclotron resonance heating; the instability of anisotropy driven modes (Weibel-type, whistler, mirror, firehose instabilities, etc.); the shear driven generation of pressure anisotropy, which plays an important role in kinetic heating in turbulence (e.g. in the solar wind) or in the loss of conservation of the magnetic moment; the role of anisotropic pressure profiles in the onset of Alfvenic modes in tokamaks; the role of pressure anisotropy in magnetic reconnection; the role of anisotropic pressure profiles for the onset of fluid shear instabilities like the Kelvin-Helmholtz at the flanks of the magnetopause; the role of pressure anisotropy in the gyroviscous closures, or in the modelling of FLR effects for magnetoacourstic modes; etc.). I do not say to mention all of these points but.. your discussion seems to me to be quite limited, with respect to them, especially considering that your analysis seems to me to neglect pressure anisotropy effects (you say to use Maxwellian equilibrium distributions).
- Section 2, Eq. 90: You justify the neglect of the displacement current in Ampere’s law with the steadiness hypothesis (line 96). Please rephrase and better motivate this choice, since, if with steadiness you mean the simple neglect of the partial time derivative, this is in my opinion inconsistent with the further linear mode analysis in which you retain the time evolution of perturbations.
- Section 2, line 158: If I am not wrong \lamda_D is not defined.
- Section 2, line 177-187: I suggest to revise these lines so to make their reading easier. You introduce the variable \chi at line 177 by relating d\chi to dl, but the definition of \chi is given only at line 182. Also note that at line 185 you make reference to the rewriting of one equation (30) which appears only several lines below (line 198).
- Section 3, line 236: I suggest to put some moduli, since formally you are defining \eta as the “ratio” of two vectors (i.e. the gradient vectors of T and of n).
Round 2
Reviewer 1 Report
Please see attached file.

Some minor issues have appeared in the last version.
Author Response
Answers to the comments of Reviewer #1:
This manuscript has been greatly improved with respect to the previous version, and most issues have been properly addressed. However, there are a few details that the authors should consider. Numbers refer to my comments to the previous version.
- In the modified manuscript, the authors have written “it’s position” instead of “its position”.
Reply:
Thanks for the reviewer to point it out! We have changed the original text according to the suggestion. Therefore, the expression “ and change with it’s position along the magnetic field line” is changed to be “ and change with its position along the magnetic field line” in the revised version of the manuscript.
- Is the ϵ mentioned here equal to the ε previously mentioned? Notation should be consistent.
Reply:
Thanks for pointing it out! They are in the same meaning, and we use the symbol ϵ uniformly in the revised version of the manuscript.
- “Other kinds of planets may not hold such kind of magnetic field configuration”. This is not enough explanation. For instance, Jupiter’s field can also be regarded as a dipole, so the specification of terrestrial planets needs a better justification, or not be mentioned.
Besides, in page 6, last paragraph, it says “instated”. Should it be “instead”. And in page 7, first paragraph, it seems it should be “which was not noticed before”, or “which has not been noticed before”.
Reply:
Thanks for pointing it out! As stated by the reviewer, there are other non-terrestrial planets that also have dipole field patterns, so we will not mention them. Correspondingly, the expression “Further investigations should be taken on the entropy mode with dipole magnetic confined plasma to reveal the characteristics of the entropy mode in magnetospheric plasmas of the terrestrial planets” has been changed to be “Further investigations should be taken on the entropy mode with dipole magnetic confined plasma to reveal the characteristics of the entropy mode in magnetospheric plasmas” in the revised version of the manuscript.
Besides, the corresponding spelling and grammar errors were also corrected in the revised version of the manuscript.
We hope that the Referee’s request is fulfilled.
With the best regards,
Zhibin Wang
Reviewer 3 Report
Dear Authors,
I thank you for your detailed replies and I acknowledge the amendments to the text. It seems to me clearer and more coherent now.
I have just two comments.
1) One is related to the sentence you have added at lines 63-66: "The perturbations of both the plasma density and the temperature, and the total pressure perturbation of both the ion and the electron is zero in the leading order, which leads to the changes the specific entropy of the plasmas."
I think this could be a little expanded, and the motivation for which this kind of interchange mode is named "entropy mode" could be better explained to the reader. I mean: strictly speaking, a gaseous system with uniform density, pressure and temperature does not necessarily sees its entropy change, so the point which in my opinion deserves to be recalled, too, is how the internal energy of the plasma is affected by these modes.
Then, maybe that a recall on the notion of entropy/negentropy and of information may help to connect your work to the Special Issue purposes, but in this regard I leave of course to the Editors, to discuss this point with you.
2) The second comment is related to your answer to point 8 of my previous report: I now better understand your choice and I agree with you, that the gradients you speak of are not vector quantities. But then I suggest you to specify in Fig. 3 that you use the symbol "\nabla" just to express a directional derivative (or, maybe, in order to avoid ambiguity, you might even replace \nabla with \partial_psi, or something similar, for example), since the symbol "nabla" of a function is usually used to express its gradient in a vector sense.
Except for these minor things, I am finally glad to express my favorable opinion for acceptance of your paper on Entropy: in spite of the fact that I suggested below "accept after minor revision", I do not need to see the manuscript again. So, in order to speed up the editorial process, I would leave to the Editors to judge whether the latest points I raised have been accounted for in a satisfactorily way for them, and then to conclude the peer review process with you.
Author Response
Answers to the comments of Reviewer #3:
I thank you for your detailed replies and I acknowledge the amendments to the text. It seems to me clearer and more coherent now.
I have just two comments.
1) One is related to the sentence you have added at lines 63-66: "The perturbations of both the plasma density and the temperature, and the total pressure perturbation of both the ion and the electron is zero in the leading order, which leads to the changes the specific entropy of the plasmas."
I think this could be a little expanded, and the motivation for which this kind of interchange mode is named "entropy mode" could be better explained to the reader. I mean: strictly speaking, a gaseous system with uniform density, pressure and temperature does not necessarily sees its entropy change, so the point which in my opinion deserves to be recalled, too, is how the internal energy of the plasma is affected by these modes.
Then, maybe that a recall on the notion of entropy/negentropy and of information may help to connect your work to the Special Issue purposes, but in this regard I leave of course to the Editors, to discuss this point with you.
Reply:
Thanks for your suggestion! When the interchange mode in the plasma is considered, the plasma and the flux tube move together, the inside flux tube and outside flux tube exchange their positions, as well as the plasma contained in the flux tube. The interchange modes are usually treated as ideal MHD, and adiabatic equation of state pvγ=c is generally adopted. In this case, the entropy of the plasma does not change while exchanging their positions. The free energy (internal energy) is derived from the total pressure gradient of the plasma. For the entropy mode studied in this paper, the electrons and the ions propagate in opposite directions, and each has its own density disturbance, temperature disturbance and pressure disturbance. However, there is no total pressure disturbance in the leading order, resulting in the changes of the system entropy, which is very interesting. The free energy (internal energy) also comes from the density gradient and temperature gradient, rather than the total pressure gradient. The corresponding content is described in the manuscript as:
The entropy mode was first studied by Kadomtsev in 1960 [16]. When the interchange mode in the plasma is considered, the plasma and the flux tube move together, the inside flux tube and outside flux tube exchange their positions, as well as the plasma contained in the flux tube. The interchange modes are usually treated as ideal MHD, and adiabatic equation of state pvγ=c is generally adopted. In this case, the entropy of the plasma does not change while exchanging their positions. The free energy is derived from the total pressure gradient of the plasma. For the entropy mode studied in this paper, the electrons and the ions propagate in opposite directions, and each has its own density disturbance, temperature disturbance and pressure disturbance. However, there is no total pressure disturbance in the leading order, which leads to the changes the specific entropy of the plasmas. The driven source of this mode can be the density and/or temperature gradients, rather than the total pressure gradient.
2) The second comment is related to your answer to point 8 of my previous report: I now better understand your choice and I agree with you, that the gradients you speak of are not vector quantities. But then I suggest you to specify in Fig. 3 that you use the symbol "\nabla" just to express a directional derivative (or, maybe, in order to avoid ambiguity, you might even replace \nabla with \partial_psi, or something similar, for example), since the symbol "nabla" of a function is usually used to express its gradient in a vector sense.
Reply:
We have made corresponding changes in the original text as η=Ln/LT.
Except for these minor things, I am finally glad to express my favorable opinion for acceptance of your paper on Entropy: in spite of the fact that I suggested below "accept after minor revision", I do not need to see the manuscript again. So, in order to speed up the editorial process, I would leave to the Editors to judge whether the latest points I raised have been accounted for in a satisfactorily way for them, and then to conclude the peer review process with you.